# Hemoglobin Endocytosis and Intracellular Trafficking: A Novel Way of Heme Acquisition by *Leishmania*

**DOI:** 10.3390/pathogens11050585

**Published:** 2022-05-16

**Authors:** Irshad Ansari, Rituparna Basak, Amitabha Mukhopadhyay

**Affiliations:** Kusuma School of Biological Sciences, Indian Institute of Technology, Hauz Khas, New Delhi 110016, India; irshad.ansari@bioschool.iitd.ac.in (I.A.); rituparna.basak@bioschool.iitd.ac.in (R.B.)

**Keywords:** hemoglobin, heme, endocytosis, traffick, Rab GTPase, *Leishmania*

## Abstract

*Leishmania* species are causative agents of human leishmaniasis, affecting 12 million people annually. Drugs available for leishmaniasis are toxic, and no vaccine is available. Thus, the major thrust is to identify new therapeutic targets. *Leishmania* is an auxotroph for heme and must acquire heme from the host for its survival. Thus, the major focus has been to understand the heme acquisition process by the parasites in the last few decades. It is conceivable that the parasite is possibly obtaining heme from host hemoprotein, as free heme is not available in the host. Current understanding indicates that *Leishmania* internalizes hemoglobin (Hb) through a specific receptor by a clathrin-mediated endocytic process and targets it to the parasite lysosomes via the Rab5 and Rab7 regulated endocytic pathway, where it is degraded to generate intracellular heme that is used by the parasite. Subsequently, intra-lysosomal heme is initially transported to the cytosol and is finally delivered to the mitochondria via different heme transporters. Studies using different null mutant parasites showed that these receptors and transporters are essential for the survival of the parasite. Thus, the heme acquisition process in *Leishmania* may be exploited for the development of novel therapeutics.

## 1. Introduction

Leishmaniasis is one of the most devastating tropical diseases, which is a threat to more than 0.7–1 million people in 100 endemic countries worldwide [1]. The disease is caused by 20 different species of the unicellular protozoan parasite, *Leishmania* [2]. The parasite has two life forms; one is the flagellated promastigote form, which inhabits the midgut of the phlebotomine sandfly, and the other is the non-motile amastigote form, which resides in a mammalian host. Leishmaniasis has been mainly classified into four clinical subtypes based on the symptoms: visceral leishmaniasis (kala-azar), cutaneous leishmaniasis, mucocutaneous leishmaniasis and post-kala-azar dermal leishmaniasis (PKDL). In cutaneous leishmaniasis, a skin ulcer forms at the site of an infected sandfly bite. It is mostly self-curable, but the time of healing varies depending on the *Leishmania* species and infected individuals. *L. mexicana, L. amazonensis* and *L. tropica* are the major causative agents of cutaneous leishmaniasis. Mucocutaneous leishmaniasis manifests as an infection in the mucosal membrane, which leads to progressive destructive ulcers in the mucosa, spreading from the mouth and nose to the larynx and pharynx. It is predominantly caused by *L. braziliensis*. Visceral leishmaniasis (VL) is the most severe form of the disease, which is caused by *L. donovani*, *L. infantum* and *L. chagasi*. It is a systemic disease that is deadly if left untreated [2,3]. For more than 70 years, the antimonial drugs meglumine antimoniate and sodium stibogluconate have been used for the treatment of leishmaniasis [4]. Antimonial drugs are toxic and have adverse reactions, including acute pancreatitis and cardiac arrhythmia [2]. Due to the high toxicity of antimonial drugs and the appearance of drug resistance, Amphotericin B, Miltefosine and Paromomycin are used as alternative anti-leishmaniasis drugs [5,6,7]. However, each of them has severe side effects, which include hypokalemia, nephrotoxicity and hepatic transaminitis [8,9]. In addition, there is no appropriate vaccine available for leishmaniasis [10]. Therefore, there is a need to develop better chemotherapeutic agents against leishmaniasis. Understanding the biology of *Leishmania* with respect to growth, differentiation and key intracellular processes may help in the identification of novel molecular targets for therapeutic intervention of leishmaniasis. One of the best ways to identify new drugs for the disease is to exploit the biochemical difference between the host and the parasites. *Leishmania* lacks a heme biosynthetic pathway and depends on an exogenous supply of heme for its survival, presumably from the host cells. Thus, understanding the heme acquisition processes by the parasite is an attractive area to identify new therapeutic targets against leishmaniasis.

## 2. Heme Acquisition: An Essential Process for Parasite

Heme is an essential cofactor for various biological processes such as electron transport chain, oxygen transport and storage (hemoglobin and myoglobin), metabolism of drugs, transcriptional regulation and signal transduction. It is an iron-containing porphyrin (protoporphyrin IX), in which most of the porphyrins contain iron as a central molecule. The central iron molecule can be oxidized into the ferric (Fe^+3^) or ferrous (Fe^+2^) oxidative state [11]. Different types of heme are present, such as heme A, heme B and heme C; the most common heme is heme B. Heme B is present in hemoglobin, myoglobin and other heme proteins. Heme is vital for most organisms such as archaea, bacteria and eukaryotes. In photosynthetic eukaryotes, heme biosynthesis initiates from the binding of glutamate to tRNA^Glu^. However, in heterotrophic eukaryotes, heme synthesis starts with the condensation of glycine with succinyl Co-A [12]. The rest of the seven steps in the heme biosynthetic pathway are evolutionarily conserved in most organisms. In contrast, the heme biosynthetic pathway is fully or partially absent in flagellated kinetoplastid parasites. For example, *Leishmania* can perform only the last three steps of the heme biosynthetic pathway in mitochondria, which involves coproporphyrinogen oxidase, protoporphyrinogen oxidase and ferrochelatase [13]. However, *Trypanosoma* lacks a complete heme biosynthetic pathway [14]. Therefore, these trypanosomatid parasites need to scavenge heme from their host for their survival.

## 3. Hemoglobin: Potential Source of Heme

As *Leishmania* is an auxotroph for heme, parasites require heme or pre-formed porphyrins for their growth from the extracellular milieu [15] and possibly rely on the host cell for the supply of heme [16]. However, free heme is toxic as it produces reactive oxygen species and damages the cell membrane [17]. Therefore, free heme/iron is almost not available in the host cells. Most of the heme in the cells is present in a complex with iron-binding proteins such as transferrin, lactoferrin, hemoglobin and plasma ferritin [18]. Incidentally, mammalian cells also import iron by endocytosis of iron-bound transferrin via the transferrin receptor [19], whereas hemoglobin is endocytosed via the CD163 receptor on macrophages [20]. Similarly, many bacterial [21,22] and fungal [23] pathogens have evolved various mechanisms to uptake host hemoproteins and extract heme from these molecules to meet their requirement. Presumably, *Leishmania* also salvages heme from macrophage hemoproteins such as Hb. This is well supported in the life cycle of *Leishmania* (Figure 1), as both amastigotes in the mammalian host and promastigotes in the insect vector have access to host Hb. The intracellular amastigote form of *Leishmania* resides in the macrophages, which take senescent RBC [24] and endocytose Hb by CD163 [20]. Interestingly, it was also shown that *Leishmania* infected macrophages phagocytose of more RBCs, possibly to acquire heme from the Hb present in erythrocytes [25]. Similarly, promastigote form in the insect gut has access to Hb from the lysed RBC in the sandfly gut [26]. This is supported by the fact that *Leishmania* can be grown in vitro in a serum-free culture medium containing blood lysate [27], and the blood lysate substituted with Hb in the culture medium renders them to grow even better [28]. In addition, the heme requirement by the parasite is met by the addition of heme directly into in vitro culture medium [29] for parasite growth. Since the free heme is not present in the bloodstream of the host and the fact that Hb can be used as a heme source at least in vitro, it is possible that *Leishmania* parasites facilitate the uptake of host cell hemoproteins to salvage heme by intracellular degradation of internalized hemoprotein. Thus, the presence of some proteins/receptors mediating the Hb endocytosis is crucial for the growth of the parasites. Therefore, it is necessary to understand the mechanism of endocytosis and intracellular trafficking of Hb to the appropriate destinations in the parasite. Here, we briefly summarized the intracellular trafficking processes in mammalian cells and compared similar processes in *Leishmania* to understand the regulation of the Hb trafficking in the parasite.

## 4. Endocytosis and Intracellular Trafficking in Mammalian Cells

Endocytosis is a process by which cells internalize various molecules from the extracellular milieu and play a key role in different processes such as nutrient uptake, antigen presentation and maintenance of cellular homeostasis. There are various modes of endocytosis, such as phagocytosis, macropinocytosis, clathrin-mediated endocytosis, caveolae-mediated endocytosis and clathrin or caveolae independent endocytosis [30]. After binding to the cell surface receptor, internalized cargoes are trafficked to the early endosomes, and subsequently, they are sorted to different intracellular destinations. The transfer of cargo from the donor membrane to the acceptor vesicle occurs through a coordinated series of vesicle fusion events specifically regulated by Rab GTPases and SNARE proteins [31]. Fusion between two particular compartments is regulated by a specific Rab along with cognate SNAREs to ensure proper targeting of the cargo to the appropriate intracellular destination.

The intracellular trafficking pathways can be broadly divided into endocytic pathways and secretory pathways. In the endocytic pathway, cells internalize material from the extracellular environment into the cell by binding with a specific cell surface receptor. Subsequently, ligands are internalized into early endosomes, where they are sorted to different destinations depending on the receptor system. Finally, ligands are trafficked to the lysosomes via late endosomes, where the cargo is degraded by the hydrolases present in the low pH of the lysosome. The degradation of the cargo in the lysosomes generates essential molecule, which is utilized by the cells depending on the type of ligand. The secretory pathway or exocytic pathway is involved in the trafficking of newly synthesized proteins, lipids and carbohydrates to various locations such as plasma membrane, endo-lysosomal system, mitochondria and the secretion of molecules outside the cell. It is a highly organized process that includes various steps involving post-translational modifications, packaging and targeting of newly synthesized proteins to their specific locations. Most of the newly synthesized proteins that are synthesized in the rough endoplasmic reticulum or in the cytosolic ribosomes follow the conventional secretory pathway via the trafficking of proteins through the ER-Golgi network to its target location. Any deviation from the classical/conventional pathway is termed as a non-conventional or uncanonical secretory pathway.

Rab GTPases are master regulators of vesicle trafficking, as evident from studies in mammalian cells. They regulate vesicle motility and membrane fusion through the recruitment of specific SNARE proteins and various effector molecules [31]. There are about 70 Rab GTPases in humans that are localized on distinct membrane compartments (Figure 2) and regulate the vesicular trafficking in the appropriate direction [32]. In the endocytic pathway, Rab5 localizes on the early endosomes and mediates the transport from the cell surface to early endosomes [33,34]. In contrast, Rab7 localizes in the late endosomes and regulates the transport of cargo from the early to the late endosome/lysosomes [35,36]. Rab4 involves the fast recycling of surface proteins from early endosomes, and Rab11 regulates the slow recycling of cargoes from endocytic recycling compartment [34,37]. Rab9 and Rab24 are also localized on the late endosomes. Rab9 mediates the trafficking of lysosomal enzymes from TGN to the lysosome [32,35]. In the secretory pathway, Rab1 regulates the anterograde transport from ER to Golgi whereas Rab2 mediates the retrograde transport from the Golgi to the ER. In addition, intra-Golgi transport is regulated by several Rabs, namely Rab6, Rab12, Rab13, Rab33, etc. Conversely, Rab3 and Rab8 regulate the transport of secretory protein from the Golgi to the extracellular milieu [31,38].

SNAREs are another group of coiled-coil proteins that confer specificity to vesicular fusion events in association with Rab GTPases, and approximately 36 members of this family of proteins are reported in mammalian cells [39]. Similar to Rab GTPases, specific SNARE also localizes on distinct intracellular compartments (Figure 3). In the endocytic pathway, Syntaxin13 localizes on the early endosomes and regulates trafficking from the early endosomes to the recycling endosomes, whereas early endosomal Syntaxin8 controls trafficking between early compartments. In contrast, Syntaxin7 plays a major role in regulating trafficking from the endosomes to the lysosomes [40,41]. Thus, the internalization of the ligands and transport to the lysosomal compartments is regulated by several endocytic Rabs and SNAREs, whereas transport of the newly synthesized receptor to the cell surface is controlled by secretory Rabs and SNAREs. However, intracellular trafficking in unicellular parasites is not well documented, apart from a few reports in Trypanosomatid parasites. Thus, it is important to characterize the intracellular trafficking pathway in *Leishmania* to understand the regulation of Hb endocytosis in the parasite.

## 5. Intracellular Trafficking in *Leishmania*

*Leishmania* is a protozoan parasite, and intracellular trafficking in this parasite is not yet characterized. However, recent reports emerging from *Trypanosoma* indicate that some of the Rabs and SNAREs are also conserved in this parasite [42,43]. Simultaneously, we also identified and characterized several Rabs in *Leishmania* to understand the intracellular trafficking pathways in the parasite.

*Leishmania* expresses a wide range of surface proteins to form protective receptors for nutrient uptake. *Leishmania* forms a protective coat on its surface known as glycocalyx, which is composed of lipophosphoglycan (LPG), proteophosphoglycan, glycophosphatidyl inositol (GPI) anchored proteins, free GPI glycolipids and glycoinositol phospholipids [44,45]. *Leishmania* also secretes several virulence factors into the host cells, such as gp63 (a metalloprotease), cysteine protease and LPG, which modulate the host cell signaling machinery for the survival of the intracellular parasite [46,47]. Among them, Ldgp63 is the major virulence factor [48,49]. Both endocytosis and exocytosis occur through the flagellar pocket of *Leishmania,* but the regulation of endocytic and secretory pathways are not characterized in the parasite. We initiated the studies to determine the role of different Rab GTPases in the regulation of intracellular trafficking in *Leishmania*.

We showed that *Leishmania* has a well-developed clathrin-mediated endocytic pathway such as mammalian cells [50]. The endocytic ligand binds to the receptor and recruits clathrin via the adaptor to form coated vesicles in *Leishmania*. Finally, the coated vesicle is pinched off from the plasma membrane by dynamin homologs in the parasite (unpublished data). We cloned and expressed early endosomal Rab5 from *Leishmania* and showed that Rab5 regulates the homotypic fusion between early endosomes [51]. Subsequently, we found that *Leishmania donovani* has two isoforms of LdRab5 (LdRab5a and LdRab5b), and both of them are localized in the early endocytic compartment. LdRab5a regulates fluid-phase endocytosis, whereas LdRab5b controls receptor-mediated endocytosis in the parasite [52]. The function of LdRab5b is essential for the parasites as Rab5b knockout parasites are unable to survive in the macrophages, probably due to the unavailability of the heme from Hb [53]. We showed that Rab7 in *Leishmania* localizes in the late endosomes and mediates the transport of cargo from the early to the late/lysosomal compartment, such as mammalian cells [54]. *Leishmania* also has a well-conserved recycling pathway, which is mediated by Rab4 and Rab11 homologs in the parasite (unpublished data).

We also analyzed the secretory pathway in the parasite, as this route is important for the secretion of virulence factors and the transport of newly synthesized receptors to the cell surface. Leishmania has a well-conserved Rab1 homolog that localizes in the Golgi. Our results showed that overexpression of the dominant negative mutant of LdRab1 blocks the secretion of secretory acid phosphatase and Ldgp63 [55]. We also found that newly synthesized proteins such as gp63 exit from the ER through a well-defined COPII complex consisting of LdSar1 GTPase, LdSec23, LdSec24 LdSec31 and LdSec13. Ldgp63-containing COPII vesicle budding from the ER is inhibited by LdSar1:T34N, a dominant negative mutant of LdSar1, and consequently blocks Ldgp63 trafficking and secretion in *Leishmania* [56]. These results suggest that *Leishmania* has a conventional secretory pathway such as mammalian cells. Bioinformatics analysis predicts that *Leishmania* genomes have several putative Rabs (Table 1) and SNARE [57] domain-containing proteins (Table 2), which need to be functionally characterized to understand the intracellular trafficking pathways in the parasite.

## 6. Heme Acquisition Processes in *Leishmania*

*Leishmania* is an auxotroph for several essential nutrients such as heme, folate, etc. They need to import these essential molecules from the extracellular milieu or from the host cells for their survival. Thus, identification of the processes by which parasites meet the requirement of these essential nutrients is of paramount importance for the development of novel therapeutic strategies. However, free heme is not available in the host cells under normal physiological conditions, and it is only present as a complex with several iron-binding proteins such as hemoglobin, transferrin, lactoferrin, ferritin, etc. Therefore, it is possible that the parasite may acquire heme/iron-containing proteins by endocytosis via the expression of a specific receptor or transporter on the parasite surface. Subsequently, various receptors/transporters were identified in *Leishmania,* which mediate endocytosis of these molecules [58]. For example, an LDL receptor-like molecule was identified on the surface of *Leishmania,* possibly for scavenging cholesterol and lipids from LDL particles; however, the LDL endocytic process is not properly characterized in the parasite [59]. Trypanosomatid parasites are also unable to synthesize purines de novo and therefore must acquire these nutrients from their hosts. Consequently, purine transporters such as LdNT1 and LdNT2 in *L. donovani* [60,61] and NT3 and NT4 are identified in *L. major* [62]. *Leishmania* is also a folate auxotroph and thus depends on the uptake of folate from the environment. It was shown that *Leishmania* has more than one active folate transporter. Among these, FT5 is a very high-affinity folate transporter in the parasite mediating the imports of folates and related molecules under varying conditions [63]. These high-affinity active transporters enable the parasites to efficiently compete with their hosts for the acquisition of these nutrients. Thus, the parasite expresses a plethora of transporters and receptors and follows different ways to import essential nutrients from their hosts. However, the heme/iron uptake process in *Leishmania* was not well characterized until recently. The following sections focus on recently identified different mechanisms that are involved in heme/iron acquisition in *Leishmania*.

### 6.1. Transferrin: As a Source of Iron by Leishmania

It is well characterized that mammalian cells import iron via the transferrin (Tf) receptor [19]. Iron-loaded Tf (holo Tf) binds with specific high-affinity receptors (TfR) on the surface of the mammalian cells and subsequently internalizes into the early endosomes via the clathrin-mediated process. In the acidified endosomal compartment, the Tf-TfR complex releases ferric iron and Tf without Fe (apoTf) along with the bound receptor, which recycles back to the plasma membrane via endocytic recycling compartments [64]. Therefore, studies were initiated to understand the role of transferrin in the iron uptake by *Leishmania*. Interestingly, a 70 kDa transferrin binding protein was detected in *L. infantum* and *L. major* cell lysates, and a monoclonal antibody against this protein recognizes human TfR, indicating that the putative protein is potentially a transferrin receptor in *Leishmania* [65]. Subsequently, the protein is purified from *Leishmania* promastigotes using affinity chromatography on a transferrin-Sepharose column. Further characterization reveals that the *Leishmania* transferrin receptor is an integral membrane glycoprotein of a single 70-kDa polypeptide, unlike the disulfide-linked dimer of the mammalian TfR. This Tf binding protein is uniformly distributed on the surface of the parasite; however, the role of this protein in transferrin uptake by *Leishmania* is not properly documented [66]. However, it is possible that the intracellular amastigote form acquires iron from the human transferrin as it was shown that *L. mexicana* containing parasitophorous vacuoles (PV) in macrophages fuse with the transferrin receptor-positive early endosomal compartment 10 days after infection and blocks the TfR receptor recycling in the macrophages. Thus, *Leishmania* amastigotes presumably recruit TfR on PV and import iron via human Tf through the flagellar pocket [67,68]. However, transferrin endocytosis in *Leishmania* needs to be further characterized.

### 6.2. Lactoferrin: As a Source of Iron by Leishmania

Apart from transferrin, parasites may also obtain iron from lactoferrin, a member of the transferrin family of proteins that are found in most of the biological fluids in mammals. Surprisingly, lactoferrin binding protein is identified as glyceraldehyde-3-phosphate dehydrogenase (GAPDH) in mammalian cells, which also binds transferrin [69,70]. It was shown that *L. chagasi* promastigotes are able to take up ^59^Fe-lactoferrin more efficiently than the ^59^Fe-transferrin. This uptake process is saturable and is inhibited by the addition of 10-fold excess of unlabeled ferrilactoferrin, suggesting the existence of a putative lactoferrin receptor on the surface of *L. chagasi* promastigotes. Moreover, the parasite is efficiently grown in vitro in iron-depleted serum containing minimal essential medium supplemented with lactoferrin, indicating the acquisition of iron by *Leishmania* from lactoferrin [71]. Subsequent studies showed that lactoferrin also binds to the similar 70-kDa protein in promastigote, which is initially identified as transferrin binding protein in *Leishmania*. Thus, this binding site is not specific to any ferroprotein. It is possible that promastigotes preferentially take up iron in a reduced form rather than an oxidized form, as it was shown that promastigotes have NADPH-dependent iron reductase activity, which is discussed in detail in the heme/iron transporters section. Thus, extracellular iron is possibly reduced prior to its internalization [72]. These results suggest that the parasite possibly utilizes various iron sources to survive in the diverse environments in the insect and the mammalian hosts.

### 6.3. Hemoglobin: As a Source of Heme by Leishmania

Hemoglobin is the major heme carrier protein in the mammalian cells, which is sequestered in RBCs. As aged RBCs are phagocytosed by macrophages and are degraded in the phagolysosomal compartment in macrophages where *Leishmania* resides, the parasite may acquire heme from the host Hb coming from degraded RBCs [73]. The first clue of *Leishmania* using Hb as a source of heme comes from the observation that parasites can be grown in vitro in a serum-free culture medium containing blood lysate [27] or Hb [28]. Therefore, it is possible that *Leishmania* may have developed a process to take up the Hb and salvage heme by intracellular degradation of internalized Hb. With this notion, we initiated studies to identify the Hb receptor (HbR) in *Leishmania*.

We showed that Hb specifically binds with a putative high-affinity binding site present in the flagellar pocket of *L. donovani* promastigotes with saturation kinetics. Subsequently, bound Hb is rapidly internalized into the parasites and is degraded in the lysosomal compartment and thereby releasing the heme from intracellular degradation of the Hb. The Hb binding with *Leishmania* is specifically competed by unlabeled Hb but not by globin or hemin or other heme- or iron-containing proteins indicating that the parasite receptor specifically recognizes Hb. Finally, a 46 kDa protein was identified as a Hb binding protein from the detergent-solubilized surface membrane preparation of *Leishmania* promastigotes using hemoglobin-agarose affinity chromatography [74]. In order to understand the role and nature of the 46-kDa protein in Hb endocytosis in the *Leishmania,* the 46-kDa protein was purified to homogeneity from *Leishmania* promastigote membrane. The purified protein specifically binds Hb and antibodies against the purified protein and blocks Hb endocytosis in promastigotes, demonstrating that the 46-kDa protein is indeed a Hb-receptor (HbR) in *Leishmania*. Consequently, the full-length HbR gene is cloned, and recombinant protein is purified. To our surprise, sequence analysis of the cloned HbR revealed the presence of hexokinase (HK) signature sequences, ATP-binding domain and PTS-II motif, indicating that the cloned protein may be a hexokinase from *Leishmania*. It is puzzling, but cell lysate prepared from HbR-overexpressing *Leishmania* promastigotes shows enhanced HK activity in comparison with the untransfected cells. Moreover, HbR-overexpressing cells bind and internalize Hb more efficiently than untransfected control parasites. In summary, these results demonstrate that HbR in *Leishmania* is a hexokinase [75]. However, these results also open up several intriguing areas for future studies. For example, how is hexokinase targeted to the flagellar pocket to bind Hb? It is possible that parasites may generate two species of hexokinase either by using different start sites or by alternative splicing? Subsequently, full-length protein is transported to glycosomes and acts as a hexokinase, whereas the truncated protein is targeted to the flagellar pocket to be a HbR as we found that PTS deleted protein is predominantly localized in the flagellar pocket (unpublished data). We are currently evaluating how it is happening within the parasite. We also tried to determine how newly synthesized HbR is transported to the flagellar pocket from the ER-Golgi network in *Leishmania*. Our results suggest that though *Leishmania* has a conventional Rab1 dependent secretory pathway, newly synthesized HbR traffic from the ER-Golgi network to the flagellar pocket follows a Rab1 independent, unconventional secretory pathway. Similarly, another open question is to determine whether hexokinase activity of HbR is needed for Hb endocytosis. However, we found that mutation in the hexokinase motif in the HbR completely abrogates the hexokinase activity of the protein, but hexokinase defective mutant HbR still binds Hb in vitro (unpublished data). It would be interesting to explore whether any other downstream accessory protein in Hb endocytosis machinery requires hexokinase activity of the HbR. 

Further characterization reveals that the N-terminus of the HbR is the extracellular Hb-binding domain, and the C-terminus is the cytoplasmic signaling domain of HbR [51,75]. Interestingly, the HbR sequence is well conserved in different species and strains of *L. amazonensis*, *L. infantum*, *L. donovani*, *L. major*, *L. turanica*, *L. gerbilli*, *L. tropica* and *L. enrietti,* with 97 to 100% sequence identity indicating that other species of *Leishmania* possibly use HbR for heme acquisition [76]. In addition, immunolocalization studies demonstrate that the HbR is also present in intracellular amastigote form [77] and mediates Hb uptake in amastigotes [53]. More recently, a 40 amino acids sequence (SSEKMKQLTMYMIHEMVEGLEGRPSTVRMLPSFVYTSDPA) in the N-terminal of HbR was identified as a Hb-binding domain [78]. This peptide completely blocks the Hb uptake in both promastigote and amastigote forms of *Leishmania* and thereby inhibits the growth of the parasite. Thus, this mimicking peptide may be used as a potential therapeutic agent against *Leishmania*.

Subsequently, studies were carried out to understand how receptor-bound Hb is internalized and is trafficked to appropriate intracellular destination to generate intracellular heme from the degradation of internalized Hb. It is well established in mammalian cells that extracellular ligands are internalized either by fluid-phase or clathrin-mediated endocytosis [79,80]. Among these modes of endocytosis, the most well-studied process is clathrin-mediated endocytosis. In the clathrin-mediated endocytosis, the cytoplasmic domain of the receptor is generally activated upon ligand binding, which interacts with endocytic adaptors, which in turn bind to the clathrin to form coated pits [81]. Finally, Dynamin GTPase cleaves the coated pits from the plasma membrane and internalizes the cargo in coated vesicles [82]. In order to understand the modes of Hb internalization in *Leishmania*, the clathrin heavy chain from *Leishmania* (Ld-CHC) is cloned and overexpressed in the parasites. Interestingly, Ld-CHC is shown to localize in the flagellar pocket of *Leishmania* along with bound Hb. Kinetic analysis of intracellular trafficking of Hb in Ld-CHC overexpressing *Leishmania* promastigotes reveals that Hb is associated with Ld-CHC coated region at early time points of internalization, and subsequently, Ld-CHC dissociates from the Hb-containing vesicles, indicating that clathrin-coating and uncoating regulate Hb trafficking in *Leishmania*. In addition, overexpression of dominant negative mutant of clathrin heavy chain of *Leishmania* (GFP-Ld-CHC-Hub) completely blocks the Hb internalization and causes severe growth defect in the parasite, demonstrating that the Hb internalization by clathrin-mediated endocytosis is essential for *Leishmania* [50]. After high-affinity binding of Hb with the HbR in clathrin coated-pits, coated vesicles are cleaved by Dynamin homologs in *Leishmania* (unpublished data) and internalize into discrete intracellular vesicles, presumably early endosome-like compartments within 5 min, as visualized by the internalization of gold conjugated Hb [74]. This is the first demonstration that Hb trafficking in *Leishmania* is probably regulated by vesicle fusion, as shown in the higher eukaryotic cell. 

As Rab GTPases are master regulators of the intracellular trafficking pathway by vesicle fusion [38], therefore endocytic Rabs such as Rab5 (LdRab5) and Rab7 (LdRab7) are cloned from *Leishmania* to understand the intracellular trafficking of Hb in the parasite. Subsequently, Hb-containing early endosomes are isolated from *Leishmania* promastigotes and determine the role of LdRab5 in homotypic fusion between endosomes in vitro to understand the trafficking of the Hb in *Leishmania*. It was shown that homotypic fusion between the early endosomes in *Leishmania* is regulated by LdRab5, whereas fusion between early endosomes with late endosomes is dependent on LdRab7 [51]. These findings suggest that the early step of Hb endocytosis is regulated by LdRab5, and transport of the Hb from the early endosome to the late endosomes/lysosome is mediated by LdRab7. In order to unequivocally prove the role of LdRab5 and LdRab7 in Hb trafficking in vivo, transgenic parasites are generated by overexpressing the LdRab5, LdRab7 or their dominant negative mutants. Incidentally, it was found that *Leishmania* has two isoforms of Rab5, namely LdRab5a and LdRab5b. LdRab5a is shown to regulate the fluid phase endocytosis, whereas LdRab5b controls the receptor-mediated endocytosis of the Hb in the parasite. Kinetic analysis of the Hb endocytosis using Alexa594 labeled Hb in promastigotes showed that Hb enters into LdRab5-positive early endosome within 5 min of internalization and is subsequently trafficked to the LysoTracker-positive lysosomes at about 45 min where Hb is degraded to generate intracellular heme as no Hb is detected at about 60 min after internalization. This process of lysosomal transport is accelerated in the transgenic parasites overexpressing LdRab5a:WT, whereas bound Hb is stuck in the flagellar pocket in the parasites overexpressing LdRab5b:N133I, a dominant negative mutant of LdRab5b [52]. Subsequently, LdRab5b^-/-^ null-mutant parasites are generated, and these parasites show severe growth defects and are unable to internalize Hb. However, these cells are rescued by exogenous addition of hemin in the growth medium, demonstrating that *Leishmania* internalizes Hb in Rab5b dependent process and generates intracellular heme by lysosomal degradation for its survival [53]. Interestingly, LdRab5b^-/-^ parasites can infect the macrophages but are unable to survive after 96 h of infection in comparison to the normal parasite. These results suggest an interesting possibility of using Rab5b null-mutant *Leishmania* as a new vaccine candidate, as these parasites can infect macrophages but are unable to survive in macrophages, and therefore, it may generate a protective immune response in an infected host. In order to understand the mechanism of trafficking of internalized Hb to lysosomes in *Leishmania*, similar studies were carried out using transgenic parasites overexpressing LdRab7 or its mutants. Interestingly, LdRab7 is shown to localize both on the early and the late endocytic compartments in *Leishmania* and thereby connecting both early and late events of Hb endocytosis. Overexpression of LdRab7:WT in *Leishmania* induces transport of internalized Hb to lysosomes, where it is rapidly degraded. In contrast, transgenic parasites overexpressing LdRab7:T21N, a GDP-locked mutant of Rab7, significantly inhibit the degradation of internalized Hb by blocking its transport to the lysosomes. Most importantly, parasites overexpressing LdRab7:T21N show growth defects possibly due to the nonavailability of heme from Hb degradation. This is supported by the fact that the addition of exogenous hemin recovers the growth of these cells almost to the control level. Thus, Rab7-mediated transport of internalized Hb to the lysosomes is required for the generation of intracellular heme from Hb degradation in *Leishmania* [54].

Thus, *Leishmania* devices a novel process to salvage heme from the Hb. In summary, *Leishmania* expresses a high-affinity specific receptor for Hb in the flagellar pocket, which internalizes bound Hb by clathrin-mediated endocytosis. Subsequently, the internalized Hb is transported to the lysosomes via early and late compartments, where it is degraded to generate intracellular heme, which is used by the parasite for its survival (Figure 4). Thus, blocking the Hb endocytosis by an appropriate inhibitor is an attractive area to explore for the development of new therapeutic molecules. In this notion, immunization with the HbR is found to induce sterile protection against the infection with virulent parasites in mice and hamsters, demonstrating that HbR is a novel vaccine candidate for visceral leishmaniasis [76]. However, it is also intriguing to understand how the heme released from the degradation of the Hb in the parasite lysosomes is trafficked to different intracellular destinations for its use. This pathway is not very well characterized but beginning to be elucidated. It emerges from recent studies that some of the heme transporters possibly regulate this process.

### 6.4. Heme/iron Transporters in Leishmania

The first heme-binding sites on the promastigote membrane were reported in *Leishmania mexicana amazonensis* [83]. It assumes to play a role in the transformation of promastigotes to amastigotes as it requires higher oxygen consumption. Analysis of *L. amazonensis* genome identified another *Leishmania* iron transporter 1 (LIT1), which is found to be present on the plasma membrane of iron-deprived promastigotes or intracellular amastigotes [84,85]. The LIT1 null mutant parasites show growth defects and are unable to form lesions in the parasite-infected animal model, indicating that this transporter plays an important role in the parasite survival. Subsequently, *Leishmania* ferric reductase 1 (LFR1) protein was identified, which is also expressed in the parasites in iron-depleted conditions and reduces the iron from the ferric (Fe^3+^) to the ferrous (Fe^2+^) state [86]. Moreover, the iron transport function of the LIT1 is dependent on LFR1, and both are overexpressed in amastigotes [87]. In addition, *Leishmania* infection triggers the production of heme oxygenase in macrophages, which catalyzes the intracellular heme to a ferrous state [88]. It is well established that ligand-bound transferrin receptor (TfR) is internalized via endocytosis and releases Fe^3+^ in acidified intracellular compartment [89]. Therefore, it is postulated that the Fe^3+^ form from the transferrin is first reduced to Fe^2+^ by LFR1, possibly in the acidified compartment in the macrophages, and then it is transported to the parasite’s cytosol by LIT1. Similarly, Nramp1, which is a cation efflux pump, can transport iron from the phagolysosome to the cytosol [90]. Iron from the cytosol is further transported to the appropriate intracellular destination for its use.

Similarly, several heme transporters were also identified in *Leishmania*. The first heme importer was identified as *Leishmania* Heme Response 1 (LHR1) in *L. mexicana amazonensis* based on the homology of *C. elegans* heme transporter HRG-4 [91]. It promotes the uptake of heme by the parasite in heme-deficient conditions and has proved to be indispensable as null mutant parasites are lethal. They also show severe defects in developing lesions in the parasite-infected mice [92]. Interestingly, LHR1 in *L. major* (LmHR1) is present in its acidic compartments and colocalizes with internalized Hb, indicating that LmHR1 possibly exports heme generated from Hb degradation in the lysosomes to the cytosol [93]. In addition, another heme transporter, LmFLVCRb, was recently identified in *L. major* promastigotes based on the homology of mammalian heme transporter [94]. Similar to LHR1, LmFLVCRb was also found to be essential for parasite survival. However, it is puzzling why *Leishmania* has two heme transporters, and both of them are essential. These open up several questions. Are they species-specific, or are they involved in the transport of the heme to different intracellular organelles? Thus, the heme derived from the Hb is transported to the cytosol by LHR1, which needs to reach the appropriate destination. Based on the role of ABC transporters such as ABCB6, which are involved in the transport of heme to the mitochondria in the mammalian cells, an ABC transporter LABCG5 with unusual topology was identified in *Leishmania.* This protein is not localized in the endocytic route in *Leishmania* and therefore may not participate in heme export from lysosomes. However, they can bind to the heme exported by the LHR1 [95]. Finally, another ABC transporter, LABCB3, was identified in *Leishmania major*, which is involved in the transport of cytosolic heme to the mitochondria and also plays a role in the mitochondrial heme biogenesis, possibly from host precursors [96]. Simultaneously, another mitochondrial iron importer, LMIT1, was identified in *L. amazonensis* based on the homology of mrs3 (yeast) and mitoferrin-1 (human) gene product. LMIT1 null mutant parasites are not viable due to strong defects in iron content and the function of mitochondria [97]. Thus, recent studies summarize how *Leishmania* is a heme auxotroph endocytose host hemoprotein and generates intracellular heme for their use (Figure 5).

## 7. Conclusions

There is now substantial evidence of how *Leishmania* meets the requirement of the heme through different receptors for hemoproteins and heme transporter. As heme is toxic and not freely available in the host cells, thus these parasites rely on host Hb to acquire heme. The other heme-binding proteins and transporters identified in *Leishmania* are possibly involved in the trafficking of the heme generated from Hb degradation to the appropriate destination. The most physiologically relevant process is to acquire heme from the host, the Hb, as it is abundantly available in the host. Thus, *Leishmania* internalizes host Hb through a specific high-affinity receptor (HbR) by clathrin-mediated endocytic process and subsequently transports it to the lysosomes via Rab5 and Rab7 dependent process, where it is degraded to generate intracellular heme. This process appears to be conserved in different species of *Leishmania* as well as in both promastigote and amastigote forms. The heme released from Hb degradation is then transported to the cytosol by different heme transporters such as LHR1 and LmFLVCRb. Finally, cytosolic heme is transported to the mitochondria by mitochondrial import proteins such as LABCB3 in *Leishmania*. This process of acquiring heme from the host Hb is also supported by the fact that visceral leishmaniasis patients are anemic and show increased erythrophagocytosis. In addition, the discovery of HbR as hexokinase opens up several interesting areas for future studies. Furthermore, apart from the fact that HbR is a novel vaccine candidate against leishmaniasis, it is also worth exploring Rab5b null mutant parasites as a vaccine candidate. Similarly, evaluation of the mimicking peptide corresponding to the Hb-binding domain is another area to explore for novel drug design against leishmaniasis. Thus, understanding the mechanism of Hb endocytosis and heme trafficking in *Leishmania* identifies several novel targets, which can be exploited for the development of therapeutic strategies against leishmaniasis.

## Figures and Tables

**Figure 1 pathogens-11-00585-f001:**
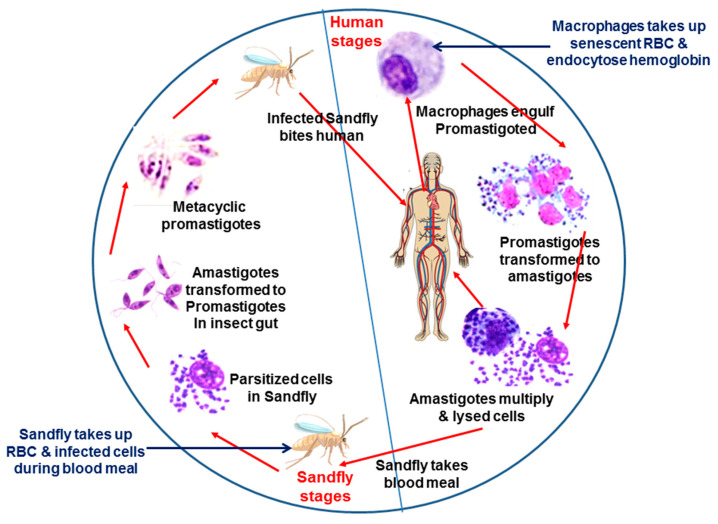
**Schematic representation shows the life cycle of *Leishmania* and source of hemoglobin.***Leishmania* has a digenetic life cycle. Transmission to humans occurs through the bite of a female Phlebotomine sandfly. In humans, promastigotes are taken up by the macrophages and transform into amastigotes in the parasitophorous vacuole (PV). Macrophages also ingest senescent RBCs, and lysed RBCs serve as source of hemoglobin for *Leishmania*. Amastigotes multiply inside the macrophages and eventually lyse cells. Parasites are released and further infect the surrounding macrophages leading to the manifestation of disease. Parasitized cells are ingested by the sandflies during the blood meal. Inside the midgut of the sandfly, amastigote transforms into motile procyclic promastigotes and colonizes their digestive tract. The insect also takes RBCs during the blood meal, which serves as a source of hemoglobin. Finally, they differentiate into infective metacyclic promastigote form and remain in the saliva of the sandfly. Parasites are transmitted to a new vertebrate host during their next blood meal.

**Figure 2 pathogens-11-00585-f002:**
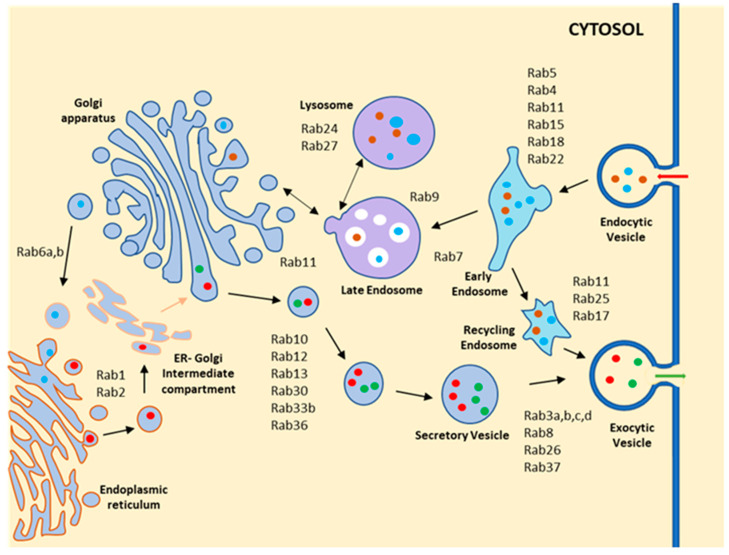
**Schematic diagram shows the localization and function of different Rab GTPases in mammalian cells.** Rab GTPases are present in distinct membranous compartments and regulate transport of cargo between the various compartments. In the endocytic pathway, Rab5 involves trafficking of the cargo from the plasma membrane to the early endosome and localizes on the early endocytic compartment. Rab4 participates in the fast-recycling pathway from the early endosome, while Rab11 and Rab35 regulate the slow recycling pathway. Rab7 helps in the trafficking of cargo to the lysosome from the early endosome, and Rab9 regulates cargo transport from the lysosome to the TGN. In the secretory pathway, Rab1 localizes on the ER Golgi intermediate compartment and regulates anterograde trafficking from ER to Golgi, whereas Rab2 involves trafficking from Golgi to ER in retrograde. Rab6 controls intra-Golgi trafficking. Rab3, Rab27, Rab8, and Rab37 regulate the transport of different secretory vesicles via the exocytic pathway.

**Figure 3 pathogens-11-00585-f003:**
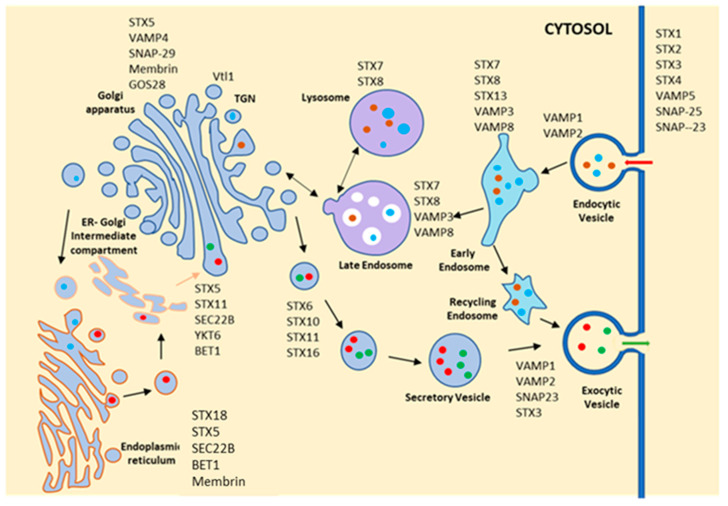
**Schematic diagram shows the localization and function of different SNAREs in mammalian cells.** Similar to Rab GTPases, SNAREs are also localized on selective compartments and provide specificity in vesicle fusion steps. They are subdivided into v-SNARE and t-SNAREs highlighted as VAMP and STX, respectively, in the figure. STX (syntaxin) 1, STX2, STX3, STX4, VAMP5, SNAP-25 and SNAP-23 are localized at the plasma membrane and possibly regulate the fusion of endocytic as well as secretory vesicles. VAMP1, VAMP2 and VAMP3, along with STX7, STX8 and STX13 are found in different endocytic compartments and regulate the different steps in trafficking of endocytic cargo from plasma membrane to lysosomes. STX11, STX5 and SEC28b are located at the Golgi apparatus, whereas STX18, Sec22b, BET1 and Membrin are found at the endoplasmic reticulum, suggesting that traffic between ER and Golgi is regulated by these proteins along with VAMP4 and SNAP29. STX6, STX10, STX11 and STX16, along with VAMP1, VAMP2 and SNAP3, regulate the transport of different secretory vesicles from the Golgi.

**Figure 4 pathogens-11-00585-f004:**
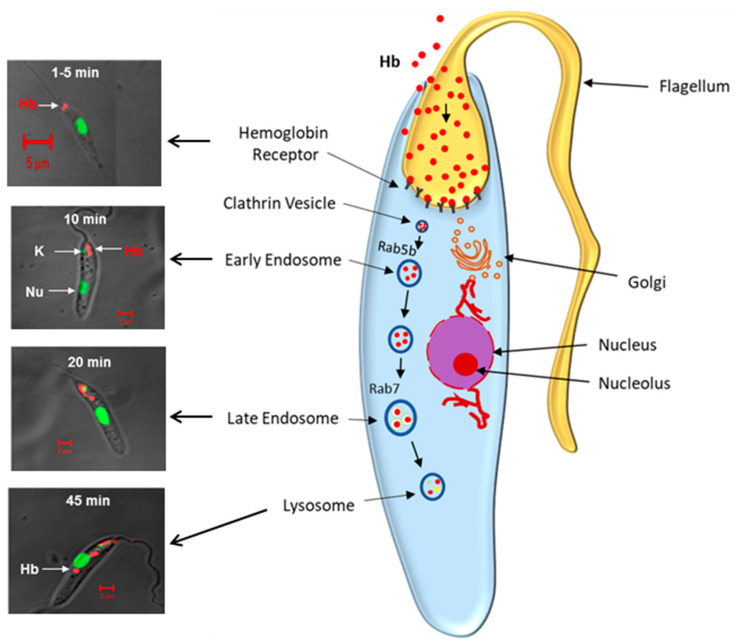
**Schematic representation demonstrates hemoglobin endocytosis in *Leishmania donovani* promastigotes.** (**Left Panel**) shows that Alexa-Red labeled Hb bind to the Hb receptor on the flagellar pocket of *Leishmania,* which traffics the Rab5 positive early endosome by 10 min and finally reaches the lysosome via late endosome in about 45 min. Nucleus is marked with green. (**Right Panel**) depicts the schematic diagram of Hb trafficking from the flagellar pocket to the lysosomes via early and late endocytic compartments in Rab5b and Rab7 dependent processes in *Leishmania* promastigotes.

**Figure 5 pathogens-11-00585-f005:**
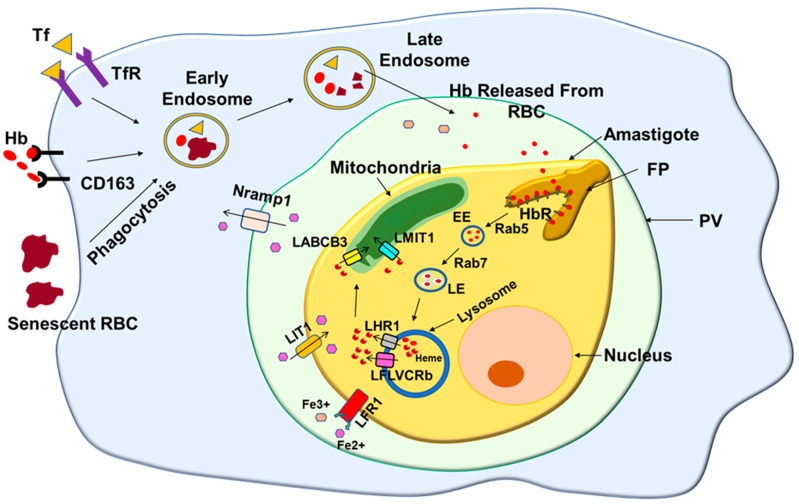
**Schematic diagram shows the heme acquisition process in *Leishmania***. Macrophages engulf senescent RBC and internalize hemoproteins such as transferrin (Tf) and hemoglobin (Hb) by specific receptor-mediated endocytic processes. These hemoproteins and RBCs are transported to the low pH compartment, where iron and hemoglobin are released from their respective hemoprotein. *Leishmania* residing in such compartment takes in the hemoglobin via their hemoglobin receptor (HbR). Subsequently, internalized Hb is transported to the parasite lysosomes, where it is degraded and releases heme. The released heme is transported to the parasite cytosol by heme transporter LHR1 and FLVCRb, which is finally transferred into mitochondria by mitochondrial heme importer LABCB3 and LMIT1. In addition, Fe^3+^ iron released by the degradation of transferrin in low acidity parasitophorous vacuole (PV) first reduces to Fe^2+^ by LFR1 and is imported by the parasite via its heme transporter LIT1.

**Table 1 pathogens-11-00585-t001:** Putative Rab homologues in *Leishmania*.

Accession Number	Putative Homologue	Location
KP308372	Rab5a	Early endosome (Ref. [51])
AY357217	Rab5b	Early endosome (Ref. [52])
EF507729	Rab7	Late endosome (Ref. [54])
KT003639	Rab1a	Golgi (Ref. [55])
CBZ12137	Rab1b	?
KY484911	Sar1	ER (Ref. [56])
CAJ08532	Rab4	?
CAJ02862	Rab11a	?
CAJ08707	Rab11b	?
CAJ08729	Rab2a	?
AYU81761.1	Rab2b	?
CBZ11916	Rab6	?
CAJ07017	Rab14	?
CAJ06306	Rab28	?
CAJ06472	Rab18	?
XP003874984	Rab23	?
CCM15425	Rab21	?
XP001681766	Rab20	?
XP003859234	Rab31	?
TPP46114	Rab35	?
CAJ08513	ARF1	?
KAG5499224.1	Rab10	?
TPP47221.1	Rab39a	?
KAG5475306.1	Small GTPase	?
KAG5503666.1	Small GTPase	?
XP001465392.1	Small GTPase	?
XP003860604.1	Small GTPase	?

Note: To identify the Rab homolog from *Leishmania*, we performed a BLAST search by using respective mouse Rab sequence as a query, which identified putative Rabs in the parasite as mentioned in Table 1. ? denotes not characterized.

**Table 2 pathogens-11-00585-t002:** Putative SNARE homologs in *Leishmania*.

Accession Number	Putative Homologue	Location
CAJ07097	Syntaxin-7	?
CBZ12338	Qa-SNARE	?
CAJ08445	Syntaxin 5	?
CAJ05451	Syntaxin-1b isoform X2	?
CAJ05450	Syntaxin-1b isoform x3	?
CAJ06328	Syntaxin-16 isoform g	?
CBZ12899	Syntaxin-16 isoform d	?
CAJ07029	Qa-SNARE	?
CAJ07013	Qa-SNARE	?
CAJ03503	Qa-SNARE	?
CAJ02128	Qb-SNARE	?
CAJ07086	Qb-SNARE	?
CAJ03721	Qb-SNARE	?
CAJ09368	Qb-SNARE	?
CAJ05002	Qb-SNARE	?
CAJ05003	Qb-SNARE	?
CAJ04850	Syntaxin 6	?
CBZ12394	BET1	?
CAJ04404	Qc-SNARE	?
CAJ04086	Qc-SNARE	?
CAJ07125	SEC22	?
CAJ02181	Qc-SNARE protein	?
CBZ12839	Synaptobrevin	?
CAJ08743	R-SNARE	?
CAJ04725	VAMP 4	?
CBZ12308	VAMP 7	?
CAJ07162	SNARE protein	?

Note: To identify the SNARE homolog from *Leishmania*, we performed a BLAST search by using respective mouse SNARE sequence as a query, which identified putative SNARE molecules in the parasite as mentioned in Table 2. ? denotes not characterized.

## Data Availability

Data are available in the cited references.

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
