# Peer review of "Hemoglobin Endocytosis and Intracellular Trafficking: A Novel Way of Heme Acquisition by Leishmania"

_pathogens, 2022, doi:10.3390/pathogens11050585_

Round 1

Reviewer 1 Report

The manuscript well summerise the knowledge on Hemoglobin endocytosis in Leishmania, as well as related background.

a few mistakes are found and list as below

1. The title seems could not fully cover the topics
e.g. the manuscript well summerised the intracellular trafficking in mammalian cell

2. Line 225-227. "Bioinformatics analysis reveals that Leishmania genomes have 30 different  Rabs and 27 putative SNARE domain containing proteins which need to be functionally characterize to understand intracellular trafficking pathway in Leishmania."
A table for Leishmania Rabs and putative SNARE with their mammalian homologs are expected.

3. Line238-249, "involve in the uptake of essential nutrients" too many other nutrients.

4. Line 355: to be consist with the following "Ld-CHC",
"clathrin from Leishmania (Ld-clathrin)" is likely to be "clathrin heavy chain from Leishmania (Ld-CHC)"

5. Line 432: typo, Leishmania mexicana amazonensis

6. Figure 5: TfnR (Is it the TfRs in line 443?), EE (early endosome), LE (late endosome)

Author Response

Point by point response of Reviewer 1.

Comment: The manuscript well summerise the knowledge on Hemoglobin endocytosis in Leishmania, as well as related background. A few mistakes are found and list as below:

Response: We thank the reviewer for appreciating our manuscript which summarise the hemoglobin endocytosis and intracellular trafficking in Leishmania.

Question 1. The title seems could not fully cover the topics e.g. the manuscript well summerised the intracellular trafficking in mammalian cell.

Reply 1. The current manuscript is focus on hemoglobin endocytosis in Leishmania. We have discussed about intracellular trafficking in mammalian cells and in Leishmania to give appropriate background. However, we have changed the title of the manuscript as suggested by the reviewer. The revised title is “Hemoglobin endocytosis and intracellular trafficking: a novel way of heme acquisition by Leishmania”.

Question 2. Line 225-227. "Bioinformatics analysis reveals that Leishmania genomes have 30 different Rabs and 27 putative SNARE domain containing proteins which need to be functionally characterize to understand intracellular trafficking pathway in Leishmania."
A table for Leishmania Rabs and putative SNARE with their mammalian homologs are expected.

Reply 2. We have incorporated new tables for putative Rabs (Table1) and SNAREs (Table2) in the revised manuscript as suggested by the reviewer.

Question 3. Line238-249, "involve in the uptake of essential nutrients" too many other nutrients.

Reply 3. We appreciate reviewer’s remark. We have modified the statement in the revised manuscript and stated as “various receptors/transporters have been identified in Leishmania to mediate endocytosis of essential molecules”.

Question 4. Line 355: to be consist with the following "Ld-CHC",
"clathrin from Leishmania (Ld-clathrin)" is likely to be "clathrin heavy chain from Leishmania (Ld-CHC)".

Reply 4. We are sorry for the mistake. We have changed it to “clathrin heavy chain from Leishmania (Ld-CHC)” in the revised manuscript as suggested by the reviewer.

Question 5. Line 432: typo, Leishmania mexicana amazonensis

Reply 5. We are sorry for typo mistake. We have stated “Leishmania mexicana amazonensis” in the revised manuscript.

Question 6. Figure 5: TfnR (Is it the TfRs in line 443?), EE (early endosome), LE (late endosome)

Reply 6. We appreciate the reviewer for point out the typo mistake. It should be TfR as suggested. We have appropriate corrected it in the revised Figure 5.

Reviewer 2 Report

Overall this manuscript covers the topic of heme usage in Leishmania satisfactorily and is well written as a review article on this topic. I have no major comment; below please find some minor comments which the authors can address during proofreading process.

L57: As Leishmania lacks… Remove ‘As’.

L86: Incidentally, mammalian cells also ‘imports’ to ‘import’

L88: No space between CD and 163.

L105: the addition ‘of’ heme

L432: Leishmania ‘Mexican’ to ‘mexicana’

L462: Are they species specific or involve’d’

L465: does not localize (locate) or is not localized (located)

Figure 1: Verbs in the figure have different tenses, some are present, some are past. Please unify.

Author Response

Point by point response to Reviewer 2.

Comment: Overall this manuscript covers the topic of heme usage in Leishmania satisfactorily and is well written as a review article on this topic. I have no major comment; below please find some minor comments which the authors can address during proofreading process.

Response: We appreciate the reviewer for finding our manuscript satisfactory and well written.

Minor comments:

L57: As Leishmania lacks… Remove ‘As’.

L86: Incidentally, mammalian cells also ‘imports’ to ‘import’

L88: No space between CD and 163.

L105: the addition ‘of’ heme

L432: Leishmania ‘Mexican’ to ‘mexicana’

L462: Are they species specific or involve’d’

L465: does not localize (locate) or is not localized (located)

Figure 1: Verbs in the figure have different tenses, some are present, some are past. Please unify.

Our reply: Thanks for point out our minor mistakes. We have made all necessary changes in the revised manuscript as advice by the reviewer.

Reviewer 3 Report

Authors: Ansari, I. et al.
Title: Hemoglobin endocytosis and intracellular trafficking: a novel way of heme acquisition by Leishmania
Ms. No.: pathogens-1668306 

This review article deals with a fascinating topic in Leishmania biology, how the parasites salvage the essential nutrient heme by internalizing hemoglobin, extracting heme, and trafficking the released porphyrin to various subcellular compartments. This is an important topic for an issue dedicated to Leishmania biology, and the authors are experts in the field, having published a substantial number of papers leading to core discoveries. 

The principal difficulty with the current version of the text concerns the need for extensive rewriting to make the language grammatically correct. Throughout the text, many definite and indefinite articles are missing, singular and plural nouns are followed by verbs of the incorrect number, present tense is often used when past tense would be correct, and there are various other problems with wording. This manuscript absolutely requires careful and extensive editing by a professional text editor or a colleague with a strong command of grammar. Following such editing, the text should become much more accessible to a wide audience and will present a scientifically important topic in a readable format appropriate for an international audience. 

I have a few specific comments listed below.

1.    Figure 1. Metacyclic promastigotes (please correct spelling) do not invade the sand fly salivary gland but rather populate the mouth parts. This is unlike other parasites such as trypanosomes and Plasmodium, which do occupy the salivary glands.
2.    Line 116 and following. There is an extensive discussion of the mammalian endocytic and secretory pathways. This material is generally relevant to the following process of hemoglobin uptake by Leishmania parasites, but it could be shortened significantly by focusing on the components most important for the latter discussion. Referring interested readers to a review article on the mammalian endocytic pathway would allow the authors to present the material more succinctly. In particular, the mammalian secretory pathway is not especially relevant to uptake of hemoglobin by the parasite, and this section could be removed.
3.    Line 208. Where a protein such as LdRab5a is first referred to, the authors should make it clear that this designation identifies the relevant protein from a particular Leishmania species, L. donovani in this case. This will be obvious to workers in the Leishmania field, but it may be confusing to those outside this discipline who may be interested in this article.
4.    Line 298. The authors refer briefly to reduction of ferric to ferrous iron and its subsequent uptake, and they come back to this point much more extensively in section 6.4. I recommend making a comment around line 298-299 indicating to readers that a more extensive discussion of this topic will follow in section 6.4. Otherwise, readers might get the impression at this point in the text that reduction of iron to its ferrous state and uptake is of minor importance, whereas it has been studied in detail.
5.    Line 327. The identification of a normally cytosolic or glycosomal protein such as hexokinase as a hemoglobin receptor is intriguing. However, it raises several puzzling and important questions that the authors should address at this point. How does a soluble intracellular protein access the external membrane of the flagellar pocket, where it must be to bind to hemoglobin? Does it potentially get secreted by a non-conventional mechanism and then bind to another protein located in the flagellar pocket membrane? Or does the protein somehow span the membrane by association with other hydrophobic proteins? The answers to these questions are probably not known, but in the context of a review, they should be presented as unsolved problems that are significant topics for future research. 
6.    Line 506. LIT1 and LMIT1 are referred to as heme transporters. They transport iron, not heme, and this sentence should be changed accordingly.

Reviewer 4 Report

This is an excellent review on the endocytic activity and intracellular trafficking in parastes of the Leishmania genus, with a special focus on the acquisition of heme that is fundamental for parasite survival and multiplication both in axenic media and inside macrophages. Te authors also present a short, but well written review, on the endocytic activity in mammalinan cells. My only suggestion, to make the review more complete, would be a short description of the same process in Trypanosoma brucei and Trypanosoma cruzi.

Round 2

Reviewer 3 Report

The authors’ answers to Comments 1-6 are fine. However, the language of the article continues to be highly problematic. I am attaching the PDF of the revised manuscript to which I have added in blue text my edits for pages 1-2. The number of such edits is far too many for a reviewer to do a comprehensive editing of the text. I really think this paper needs to be corrected by a professional editor to be of the quality of presentation that would be appropriate for publication, especially when the target audience is wide.
